# Digital Biosurveillance for Zoonotic Disease Detection in Kenya

**DOI:** 10.3390/pathogens10070783

**Published:** 2021-06-22

**Authors:** Ravikiran Keshavamurthy, Samuel M. Thumbi, Lauren E. Charles

**Affiliations:** 1Paul G. Allen School for Global Animal Health, Washington State University, Pullman, WA 99164, USA; r.keshavamurthy@wsu.edu (R.K.); thumbi.mwangi@wsu.edu (S.M.T.); 2Pacific Northwest National Laboratory, Richland, WA 99354, USA; 3Center for Epidemiological Modelling and Analysis, Institute of Tropical and Infectious Diseases, University of Nairobi, Nairobi 30197, Kenya; 4Institute of Immunology and Infection Research, University of Edinburgh, Edinburgh EH9 3FL, UK

**Keywords:** zoonosis, biosurveillance, digital surveillance, open-source information, disease taxonomy, Kenya

## Abstract

Infectious disease surveillance is crucial for early detection and situational awareness of disease outbreaks. Digital biosurveillance monitors large volumes of open-source data to flag potential health threats. This study investigates the potential of digital surveillance in the detection of the top five priority zoonotic diseases in Kenya: Rift Valley fever (RVF), anthrax, rabies, brucellosis, and trypanosomiasis. Open-source disease events reported between August 2016 and October 2020 were collected and key event-specific information was extracted using a newly developed disease event taxonomy. A total of 424 disease reports encompassing 55 unique events belonging to anthrax (43.6%), RVF (34.6%), and rabies (21.8%) were identified. Most events were first reported by news media (78.2%) followed by international health organizations (16.4%). News media reported the events 4.1 (±4.7) days faster than the official reports. There was a positive association between official reporting and RVF events (odds ratio (OR) 195.5, 95% confidence interval (CI); 24.01–4756.43, *p* < 0.001) and a negative association between official reporting and local media coverage of events (OR 0.03, 95% CI; 0.00–0.17, *p* = 0.030). This study highlights the usefulness of local news in the detection of potentially neglected zoonotic disease events and the importance of digital biosurveillance in resource-limited settings.

## 1. Introduction

Biosurveillance focuses on systematically collecting and combining health information with appropriate analysis and interpretations to achieve early warning, detection, and overall situational awareness of disease activity. Infectious disease surveillance is an important epidemiological tool used to describe the burden of disease, monitor its trends, and detect outbreaks of new and existing pathogens [1]. Traditionally, biosurveillance systems rely on the reporting of identified diseases formally by field investigators, physicians, veterinarians, laboratories, and other healthcare providers to relevant health agencies. Although traditional surveillance produces accurate and high-quality data, these systems can be expensive and require a formal health structure to operate [2]. Furthermore, these systems are hierarchical in organization, causing considerable lag in disease reporting [3]. Many times, an outbreak is underreported or unreported due to these limitations, especially in resource-limited settings [4]. 

Digital health surveillance is a form of non-traditional, internet-based, surveillance that mines large volumes of electronic data, e.g., news articles, social media, blogs, and other internet sources, to identify and monitor health-related events [5]. These digital surveillance methods act as a form of early warning systems, providing health alerts that are difficult to obtain through traditional health infrastructure. These systems utilize varying levels of automation and human scrutinization to screen large quantities of unstructured data present on the internet, filter out unnecessary data, and flag potential health threats [6]. The filtered information is then further verified by qualified experts and useful information is extracted manually or using natural language processing techniques for further analysis [3]. Early warning systems have played a critical role in informing the outbreaks of diseases, such as severe acute respiratory syndrome (SARS) and Ebola, to official health authorities [7,8]. Some of the examples of disease warning systems include ProMED-mail, HealthMap, the Global Public Health Intelligence Network (GPHIN), and the Early Warning, Alert and Response System (EWARS) [8,9,10,11]. Early warning systems are highly cost-effective, easily accessible, and provide spatial and temporal alerts of high resolution in near-real time [12]. In many low- and middle-income countries where the veterinary and public health infrastructure is rudimentary, declining, or nonexistent, digital surveillance can inform official verification, timely response, and the mobilization of health resources during disease outbreaks. Digital surveillance has been utilized by international agencies, such as the World Health Organization (WHO), as a source of epidemic intelligence in investigations. In fact, the majority of infectious disease events investigated by WHO were first identified through these informal sources, including press reports and the internet [13,14]. 

News media play an important role in the dissemination of health-related information to the general population. The local media are especially crucial in influencing public attitudes and knowledge about neglected endemic diseases that are seldom the focus of central health agencies. This type of media also serves as an important source of information concerning disease outbreaks that are not officially reported by countries, possibly due to negative ramifications on trade, travel, or tourism. With recent technological advancements, these digital health-related reports are readily available and can be used as disease intelligence for early outbreak investigation [9]. However, manually gathering, filtering, and analyzing these large amounts of unstructured data is limited by the requirement of a lot of manpower and time. To overcome this, Biofeeds, a sophisticated and powerful tool built for open-source digital biosurveillance, enabling rapid detection and enhanced analysis of emerging biological events, has been developed [15]. The tool captures information from more than 70,000 unique sources published from around the globe, including third-party aggregators, such as Google, subscription-based feeds, such as ProMED-mail, and Really Simple Syndication (RSS) feeds. Biofeeds utilizes a text analytic pipeline called Adaptable and Automated Insight Detection (AAID). The AAID pipeline has many built-in algorithms, including the ability to filter biological events based on relevance and active event confidence. 

Zoonotic diseases continue to pose major threats to human and animal health, security, and the economy. In Kenya, due to a continuous change in farming systems, preferred livestock breeds, and trading patterns, as well as close interactions between people and animals, there is a rising risk of zoonotic disease outbreaks. Though zoonotic disease surveillance is carried out by both human and animal health sectors, there is minimal integration of the surveillance systems between the two sectors, leading to underreporting of disease events [16]. Furthermore, Kenya is limited in disease surveillance infrastructure comprised of adequate diagnostic facilities across the country and appropriately trained personnel, disease reporting, and an early warning system. In this situation, a digital surveillance system has the potential to greatly advance country-wide biosurveillance efforts through automated gathering, compiling, identifying, and reporting of information related to potential new or unknown disease events from a variety of sources. Our study examined the nature and extent of zoonotic disease reporting in Kenya, the potential of digital biosurveillance in the early detection and warning of zoonotic disease events, and factors associated with the official reporting of such events. 

### Definitions 

Disease report: Open-source digital media reports including news articles, governmental and international bulletins, and other online disease warning/surveillance system notifications that report a disease occurrence, harvested through Biofeeds.Disease event: A single or group of epidemiologically related disease occurrences at a given location and time that is characterized by one or more common epidemiological indicators.Official disease event: A disease event whose occurrence was confirmed by one or more national and/or international health agencies, e.g., WHO or the World Organization for Animal Health (OIE), in the form of an official release report.Epidemiological indicators: The epidemiological information present in disease reports used to identify and describe a disease event.Disease event taxonomy: A tree-based structure of epidemiological indicators used to systematically identify and describe disease event information present in disease reports.

## 2. Results

Out of 1874 news records harvested by Biofeeds for the study period from August 2016 to October 2020, a total of 424 distinct disease reports regarding disease occurrences in Kenya were identified. The other 1450 records were either informative articles, disease warning news, scientific literature, or miscellaneous articles that included homonyms of key search terms. These records did not report ongoing disease events and hence were excluded from the study. 

Rift Valley fever (RVF) accounted for 46.7% (*n* = 198), anthrax 44.6% (*n* = 189), and rabies 8.7% (*n* = 37) of the events. No brucellosis- or trypanosomiasis-related disease events were identified in Kenya during our study period. The disease reports were further classified based on their sources of reporting as news media reports (*n* = 298; 70.3%), disease warning systems reports (*n* = 89; 21%), official health agency reports (*n* = 31; 7.3%), and Kenyan government news agency reports (*n* = 6; 1.4%). Among the news media, Nation (*n* = 35; 11.7%), The Standard (*n* = 28; 9.4%), Kenya news (*n* = 16; 5.4%), and The Star (*n* = 16; 5.4%) were the most frequently identified, all of which are domestic news media organizations. Kenya News Agency (*n* = 6) was the only Kenyan-government-run news agency. The official health agencies that reported disease events were international health organizations, including the World Organization for Animal Health (OIE) (*n* = 13; 41.9%), Food and Agricultural Organization (FAO) (*n* = 7; 22.6%), WHO (*n* = 4; 12.9%), United Nations (UN) (*n* = 4; 12.9%), and Center for Disease Control and Prevention (CDC) (*n* = 2; 6.5%). The most common disease warning systems were ProMED-mail (*n* = 42; 47.2%), Hazardous Materials Managers News (*n* = 17; 19.1%), Outbreak News Today (*n* = 15; 16.9%), and FluTrackers (*n* = 5; 5.6%). 

We gathered epidemiological indicators present in the news reports using disease event taxonomy as described in the methodology. Further, we grouped these disease reports into individual disease events using one or more common epidemiological indicators. All the detected disease events, along with their sources of reporting and various event-specific epidemiological indicators reported, are presented in Figure 1. We identified a total of 55 unique disease events belonging to anthrax (*n* = 24; 43.6%), RVF (*n* = 19; 34.6%), and rabies (*n* = 12; 21.8%). Most of the disease events were first reported by news media (*n* = 43; 78.2%), followed by official health agencies (*n* = 9; 16.4%) and the Kenyan government news agency (*n* = 3; 5.5%). Notably, of the 55 disease events, only 18 (32.7%) were ever reported by official health agencies, which included RVF (*n* = 17; 94.4%) and anthrax (*n* = 1; 5.6%), and all of them were reported by international health organizations (Figure 1). No official reports from Kenyan national health agencies were found. Furthermore, only nine events were reported both in official sources and news media. On average, news media reported these events 4.1 (±4.7) days and a maximum of 14 days faster than the official health agencies (Table 1). 

The number of officially reported disease events clustered by primary affected host group is presented in Table 2. Human cases were the main cause of event reporting for anthrax (*n* = 20; 83.3%) and rabies (*n* = 10; 83.3%); however, none of these human-related events were reported by official sources. In contrast, disease occurrences in livestock were the main cause of event reporting for RVF (*n* = 14; 73.7%) and the majority of these livestock-related occurrences were officially reported (*n* = 13; 92.9%). 

Univariate logistic regression was implemented in seven event-specific explanatory variables to study the factors associated with official reporting of disease events. Contingency table and univariate analysis of the association between the event characteristics and the official reporting are presented in Table 3. RVF events compared to anthrax had a significantly larger odds of being officially reported (odds ratio (OR) 195.50, 95% confidence interval (CI); 24.01–4756.43, *p* <0.001). There was a strong negative association between the official reporting and local media coverage of disease events (OR 0.03, 95% CI; 0.00–0.17, *p* = 0.030). In contrast, there was a significant positive association between official reporting and disease warning systems’ alerts of the events (OR 16.11, 95% CI; 2.84–305.22, *p* = 0.004). Furthermore, livestock-related outbreaks were significantly more likely to be officially reported compared to those in humans (OR 16.79, 95% CI; 4.41–78.71, *p* < 0.001).

## 3. Discussion 

Zoonotic diseases are a major concern in Kenya and their frequent occurrence causes considerable public health and veterinary concerns, and economic losses in the country [17]. Among them, anthrax, trypanosomiasis, rabies, brucellosis, and RVF have been shown to be the top five priority zoonotic diseases in Kenya [4]. All these zoonotic diseases are listed as notifiable animal diseases under OIE—Terrestrial Animal Health Code [18]. According to OIE, any outbreak related to these diseases is a subject of immediate notification to all the member nations, i.e., officially reported and released to the public through an online report [18]. Additionally, of the five zoonotic diseases, four of them (anthrax, trypanosomiasis, rabies, brucellosis) are listed as neglected zoonotic diseases by WHO [19]. Due to the lack of robust biosurveillance and reporting infrastructure in Kenya, these zoonotic disease occurrences have been considerably underreported or unreported over the years by both human and animal health agencies. Our study gathers local and international open-source data published by various news media organizations, health agencies, and surveillance systems to better understand their potential impact on advancing biosurveillance efforts. To the best of our knowledge, our research is the first country-wide study to date that investigates the role of open-source reports in the detection and situational awareness of zoonotic diseases in Kenya.

Out of all the officially reported disease events collected between August 2016 and October 2020, the majority were RVF, which had the highest odds of being officially reported. RVF is a major vector-borne transboundary disease that could threaten food security and veterinary and public health globally due to its ability to cause major and frequent epidemics. This disease has the potential to cause widespread human mortalities and morbidities as well as livestock abortions, which often leads to devastating economic impacts due to quarantine and bans on consumption of animal products and/or travel. In addition, compared to anthrax and rabies, which are endemic and easier to clinically diagnose, RVF occurs sporadically and is epidemic in nature, which quickly triggers responses from authorities, including case investigation, sample collection, and official reporting. Due to these reasons, RVF receives more attention from the global and local health community compared to the other four zoonotic diseases included in this study. An initial outbreak triggers widespread official active surveillance initiatives, especially in livestock herds, to speed up control and preventive efforts [20]. Such large-scale surveillance efforts in livestock herds during an outbreak may explain the higher odds of livestock-associated official reports compared to humans in our study.

In contrast to RVF, only one out of 24 anthrax occurrences and none of the 12 rabies occurrences that were reported by unofficial sources (news media and surveillance systems) were identified in official reports. Furthermore, brucellosis and trypanosomiasis were not reported by any of the local or international agencies. Despite the high health and socio-economic impacts, the true burden of these endemic diseases in Kenya is largely unknown [21,22,23,24]. These four diseases, like any other neglected zoonotic disease, mainly affect the poorest and most marginalized communities, particularly in low- and middle-income countries [19]. These diseases are often undersupported in terms of political profile, funding, research interest, policy, and health resources required to decrease their burden [25]. Our investigation identified many anthrax and rabies occurrences using local media sources that were not reported by official sources, further signaling severe underreporting of zoonotic disease events by official health agencies.

Our study showed that the official sources were more likely to report events related to livestock compared to humans. This finding is in agreement with a previous study carried out in Nepal where international sources, especially OIE, were more likely to report animal health events compared to human events [26]. OIE member countries have requirements for reporting unusual events related to notifiable diseases [27]. In our study, OIE was the main official health agency that reported primarily livestock-affected events. However, the majority of human cases were reported, not by official sources, but by local news media (Table 2), which could explain the negative association between official reporting and news media reporting. In addition, the majority of disease events in Kenya were first reported by news media (Figure 1). According to our study, local news media could play an important role in risk communication by timely dissemination of health information to all stakeholders in the case of a disease event. Rather than relying solely on central official sources, this critical health information could instead be obtained in near real-time directly from witnesses or lower institutions in the disease-reporting chain, such as primary health care centers or diagnostic labs. Our study underlines the potential impact that local media can have on early detection and situational awareness of zoonotic disease events when traditional biosurveillance is lacking.

There are many strengths of this research. The catchment period for this study was 50 months, which allowed us to better describe the overall status of important zoonotic disease reporting and its role in early detection and situational awareness in Kenya. We used Biofeeds, a powerful biosurveillance tool, to systematically gather disease event information from multiple sources. Furthermore, we manually read, filtered, and grouped all gathered news reports to ensure that no disease event was missed. We developed a disease taxonomy to enable systematic identification of epidemiologic information present in disease reports and to convert the unstructured text reports into a structured form for better analysis. This study lays a foundation for the automated extraction of epidemiological information present in open-source health reports with minimal human intervention.

There are a few limitations of this research. Due to the inherent problems with online retrospective analysis, sometimes only the title and description of a news article were able to be analyzed because the original webpage was no longer available. Not having access to the full article may have resulted in the exclusion of event-related information present only in the full text. In addition, although many of the events reported by unofficial sources were later confirmed by the official reports, we were not able to verify the occurrences of all the unofficial reports harvested. News media plays a critical role through timely and widespread dissemination of information related to health risks and other threats to the general public. Hence, news media reports should be accurate and reliable for effective risk communication. More research is required to better understand the accuracy of information obtained by news media and regardless, the information obtained should be verified before action is taken. Finally, we might have missed disease reports from smaller media agencies with limited to no online presence, but that may still have a strong influence and better reporting structure locally in the form of print media. With technical advancement, especially in a developing country such as Kenya, domestic sources of information will be more readily available digitally in the future, which will improve digital biosurveillance.

In conclusion, our study contributes to understanding the nature and extent of zoonotic disease reporting in Kenya. RVF was the disease most reported, followed by anthrax and rabies. Neither brucellosis nor trypanosomiasis events were ever reported, officially or unofficially, despite their huge impact and presence in Kenya. Similarly, RVF fever was the main disease reported by official sources, whereas other disease events were seldom or never mentioned. Local news was the key source for anthrax and rabies reports, indicating that local news reporting can play an important role in the early detection and situational awareness of neglected zoonotic diseases in resource-limited Kenya. Our study highlights the role of digital biosurveillance systems in reducing the time lag for early detection of disease outbreaks, their verification, and rapid mobilization of health resources to the regions of concern. Digital biosurveillance can also improve communication between local health authorities and central health agencies both at the national and global level for better assistance, especially in case of major health emergencies. With the growing public health awareness among the people, increasing availability of the internet, and universalization of digital space, there is a potential to scale such biosurveillance systems beyond the regional level to cover wider geographical areas and disease events. However, more studies are required to fully understand the role and significance of news articles in digital biosurveillance and early warning for other important infectious diseases across the world.

## 4. Methodology

### 4.1. Data Collection and Handling

Open-source disease report data related to priority zoonotic diseases of Kenya (i.e., anthrax, rabies, RVF, brucellosis, and trypanosomiasis) were collected using Biofeeds, a digital platform for open-source biosurveillance. Key search terms were generated that included location (Kenya along with all 47 Kenyan county names) and the five priority zoonotic disease names and their synonyms. A REGEX query was carried out to gather all available digital articles related to the five zoonotic disease events in Kenya published between August 2016 and October 2020 within the Biofeeds database. The search was conducted for each disease individually and then run through the AAID pipeline full-text extraction, relevance machine learning, and active event deep learning algorithms. The information, collected in JSON, included the title, description, source URL, source host, published date, full text, relevance score, and active event confidence. The pertinent disease event reports were manually identified by reading the title, description, and full text of each harvested news article.

### 4.2. Disease Event Taxonomy Construction and Report Tagging

For the systematic identification and labeling of information related to epidemiological indicators present in the disease reports, we developed a disease event taxonomy specific to the selected five priority zoonotic diseases of Kenya. This disease event taxonomy, constructed of simple hierarchical concepts, was designed to capture disease-specific epidemiological indicators present in the disease reports. Useful epidemiological information present in the title, description, and full text of the reports were manually tagged using the disease event taxonomy. This process facilitated the conversion of unstructured disease report data into a structured form for better documentation and ease of analysis. A detailed description of the disease event taxonomy used to collect these epidemiological indicators and their sub-categories is presented in Table 4.

A unique disease event was identified by using location and date of occurrence along with one or more epidemiological indicators present in the disease reports. After cross-referencing using the epidemiological indicators, all the matching reports were aggregated into unique disease events. For example, an anthrax event that hospitalized twelve people in Murang’a after consumption of infected meat was reported on 8 November 2016 21:58:10 PT in a Kenyan local news article. Seven more reports that refer to the same incident were reported in the next 11 days. All these reports were aggregated as a single event based on event location, event time, disease, infected host group, source, and mode of infection. Two reports were considered distinct if they contained information about the same event but had a unique source URL. We classified the sources of these disease reports as international/national official health agencies, news media, government news agencies, and disease warning systems. The total number of disease reports per event and details on each epidemiologic indicator were recorded. In addition, to understand the timeliness of the media platforms, the source that first reported the disease event as well as the source that first reported any epidemiologic indicator category was noted.

### 4.3. Variable Construction and Data Analysis

A logistic regression analysis was performed to understand the factors associated with the official reporting of a disease event. A disease event was considered as officially reported if its occurrence was confirmed by one or more national and/or international health agencies in the form of the disease report harvested by Biofeeds. Event-specific categorical variables used in the analysis included the different news sources that reported disease events, the country province where the event occurred, and the primary affected host group that led to event reporting. The continuous variable, number of disease reports per event, was not normally distributed and, hence, was categorized by splitting it at its tertiles (less than two, three to six, seven and more). The frequency distribution of the categorical variables describing the characteristics of the disease events were arranged in the form of contingency tables. Logic checks were conducted, and suspicious records were examined against the disease reports and corrected if required. Univariate logistic regression analysis was implemented on the contingency tables to study the association between the official reporting of the event and the event characteristics. The explanatory variables with a Fisher’s exact *p*-value of less than 0.05 were considered as statistically significant. The statistical analysis was conducted in R statistical program (R statistical package version 3.4.0, R Development Core Team [2015], http://www.r-project.org, accessed on 10 February 2021).

## Figures and Tables

**Figure 1 pathogens-10-00783-f001:**
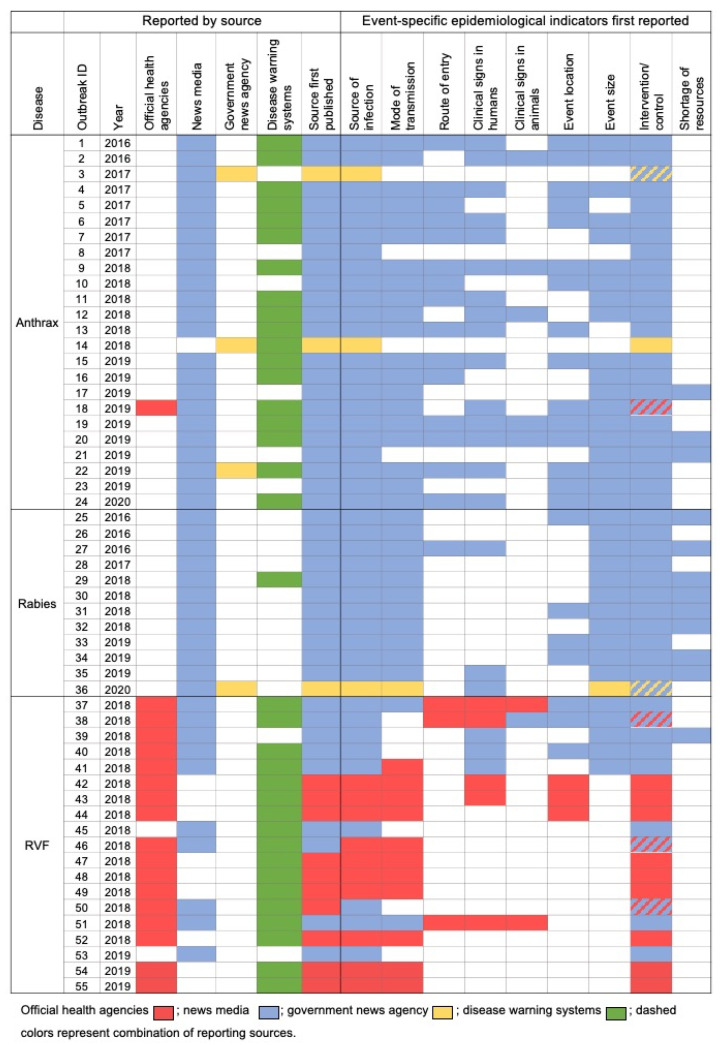
Details of disease events identified, their reporting sources, various event-specific epidemiological indicators reported, and the source that first reported them, between August 2016 and October 2020.

**Table 1 pathogens-10-00783-t001:** Details of the disease events that were reported both in news media and official health agencies along with the difference in duration in reporting between them (published online between August 2016 and October 2020).

Outbreak ID	Disease	Number of Days News Media Reported before Official Reports
18	Anthrax	3
37	RVF	2
38	RVF	2
39	RVF	8
40	RVF	1
41	RVF	5
46	RVF	14
50	RVF	−4
51	RVF	7

RVF; Rift Valley fever.

**Table 2 pathogens-10-00783-t002:** Description of official reporting of disease events published online between August 2016 and October 2020; namely, the percent of events that were officially reported out of the total identified events in the study grouped by type of disease and primarily affected host species.

Disease Events (*n* = Total Events)	Primary Affected Group Reported		
	Human (Official/Total)	Livestock (Official/Total)	Wildlife (Official/Total)
Anthrax (*n* = 24)	0% (0/20)	0% (0/3)	100% (1/1)
Rabies (*n* = 12)	0% (0/10)	0% (0/2)	0% (0/0)
RVF (*n* = 19)	80% (4/5)	93% (13/14)	0% (0/0)

**Table 3 pathogens-10-00783-t003:** Contingency table and univariate analysis of the association between the event characteristics and the official reporting of 55 disease events published in news reports between August 2016 and October 2020.

Parameters	Category	Not Reported	Reported	Odds Ratio	LCL	UCL	*p*-Value
Disease	Anthrax	23	1	Reference			<0.001
	Rabies	12	0	0.00	**	**	
	RVF	2	17	195.50	24.01	4756.43	
Government news agency reporting	Not Reported	33	18	Reference			0.291
	Reported	4	0	0.00	**	**	
News media reporting	Not Reported	1	9	Reference			0.030
	Reported	36	9	0.03	0.00	0.17	
Surveillance system reporting	Not Reported	18	1	Reference			0.004
	Reported	19	17	16.11	2.84	305.22	
Location of the event (Province)	Central	6	2	Reference			0.015
	Coast	1	2	6.00	0.38	182.92	
	Eastern	11	5	1.36	0.21	11.60	
	Nairobi	1	0	0.00	**	**	
	Northeastern	0	5	**	0.00	**	
	Nyanza	6	2	1.00	0.09	10.76	
	Rift Valley	12	2	0.50	0.05	5.01	
Primary affected group	Humans	31	4	Reference			<0.001
	Livestock	6	13	16.79	4.41	78.71	
	Wildlife	0	1	**	0.00	**	
Number of reports	Less than two	17	4	Reference			0.224
	Three to six	8	7	3.72	0.87	17.98	
	Seven and more	12	7	2.48	0.61	11.32	

LCL = 95% lower confidence limit, UCL = 95% upper confidence limit, ** inestimable.

**Table 4 pathogens-10-00783-t004:** Description of disease event taxonomy used to collect the six zoonotic disease-specific epidemiological indicators present in the disease reports published between August 2016 and October 2020.

Epidemiological Indicator	Sub-Categories
Disease	Anthrax, Rift Valley fever, rabies, brucellosis, trypanosomiasis
Species affected	Humans: children, adult, elderly Domesticated/Livestock: bovine, sheep, goat, poultry, camel, donkey, horse, dog, cat Wildlife: buffalo, rhinoceros, gazelle, giraffe, warthog, waterbuck
Source of infection	Humans to humans, animals (domestic/wildlife) to humans, environment to humans, animals to animals, environment to animals
Mode of transmission	Direct contact: infected animals, infected carcass/meat, infected animal byproducts, infected birthing fluids/placenta, animal bite Indirect contact: soil, water, air, mechanical vector, biological vector
Route of entry	Ingestion, inhalation, cutaneous/contact
Clinical presentation	General/non-specific, pulmonary, gastrointestinal, cutaneous, neurological, musculoskeletal, circulatory, reproductive, behavioral
Event location	Hospital/clinic, home residence, small farm, production farm, national park, slaughterhouse
Event size	Isolated case, single cluster (within village/town/city), multiple clusters/regional, multiple regions (counties/provinces/states), multiple countries, epidemic, pandemic
Intervention/control	Hospitalization, home isolation, movement control, passive surveillance, active surveillance, area containment/closure, travel ban, quarantine, culling, disposal, disinfection, vaccination, warning/advisory, monitoring, recall/market removal, meat inspection, vector control
Shortage of resources	Diagnostic facilities, vaccines, drugs/medication, ventilators, personal protective equipment, health care workers, misdiagnosis/medical error, delayed/no medical attention

## Data Availability

The data presented in this study are available on request from the corresponding author.

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
