# Peer review of "Digital Biosurveillance for Zoonotic Disease Detection in Kenya"

_pathogens, 2021, doi:10.3390/pathogens10070783_

Round 1

Reviewer 1 Report

In this manuscript, authors have highlighted the usefulness of digital bio-surveillance in detection of neglected zoonotic disorders. The manuscript highlights an important area of health. Zoonotic disorders in neglected and underdeveloped remote areas often go under-reported. Continued epidemiological surveillance is also not workable. Thus, this study is important in the area of neglected zoonotic diseases. Although investigators have listed the limitations of this study, digital bio-surveillance can be an additional tool to assist the ongoing transmission and epidemiological surveys in endemic areas. I have no major comments on this manuscript. 

Reviewer 2 Report

Thank you for allowing me to review the article “Digital biosurveillance for zoonotic disease detection in Kenya,” where Keshavamurthy et al present the use of digital biosurveillance as a tool for reporting incidence of zoonotic disease outbreaks in Kenya. This manuscript presents a very important and timely example of the ability of data mining to have real world impact on public health in a developing country. The manuscript is clear in its objectives and is well written.

Minor comments:  

1) The phrase “neglected endemic disease that seldom come under the radar of central health agencies…” in Lines 62-63 may confuse some readers. The phrase “under the radar” implies things unseen and I believe the authors are intending to say that the endemic diseases are seldom within the focus of central health agencies and are frequently unseen/under the radar. Suggest rephrasing to something like “…neglected endemic diseases that are seldom the focus of central health agencies.”

2) Might enhance the conclusions with stronger language about the implication of the use of this type of digital biosurveillance, like reducing lag time to official notifications and involvement. Reducing lag time to appropriating resources or treatments in areas of active outbreak. Improving communications between local health authorities and central and global health authorities that can provide assistance. Especially critical for areas of limited resources, developing countries, rural areas, food security Etc.

Reviewer 3 Report

This is a well-written study designed to understand the role digital surveillance of zoonotic diseases in Kenya. The study design and the analyses are appropriate for the questions being addressed. The insight of local news media helping to detect zoonotic disease outbreak events is an important recommendation that will be useful in future surveillance efforts. I just have a few minor suggestions to improve the manuscript.

General comments:

I think giving some more information/examples of early warning systems in the introduction would be helpful. How do these work? What constitutes an early warning system?

Could these methods be applied to other countries in Africa? What situations work best for studies like these? Or to use this type of digital surveillance? Maybe a few sentences in the discussion about broader impacts, other than just in Kenya, would be helpful.

So there were 7 univariate logistic regressions done? If so, I would state this to help the reader follow the results from the data analysis section. I would also explain in the data analysis section that the logistic regressions were done on contingency tables.

Figure 1: I think a legend explaining the colors would help readers understand the figure faster, instead of just having a description of colors in the legend.

Specific comments:

Line 52: low and middle (income) countries…

Lines 110-111: What were the other news records? There are 1,450 records that were not disease reports. What kinds of information were found, but ultimately not relevant?

Lines 138-139: As opposed to national health organizations (those in Kenya)?

Lines 348-350: Just a little confused because this seems to be repetitive, unless I am missing something. Maybe this is where you can specify that 7 univariate logistic regressions were done.

It is unclear if international health organization reports are included in Figure 1. I guess they are included as "official" and described as "official health agencies" in the legend? I would be consistent with names and terms. Similar with "disease warning systems" and "surveillance systems". These different terms are mostly confusing when referencing the figure and lines 114-118, which describe the results in the figure. I would try to use the same terms for the text and the figure legend.

Table 4 legend: disease-specific, not diseases-specific
